# Nutritional Strategies to Address Malnutrition in Dialyses Patients: A Systematic Review

**DOI:** 10.3390/nu17213478

**Published:** 2025-11-05

**Authors:** Paula Arroyo-Serrano, Rosario Alonso-Dominguez, Sebastián Mas-Fontao, Emilio Gonzalez-Parra, María Luz Sánchez-Tocino

**Affiliations:** 1Department of Nursing and Physiotherapy, University of Salamanca, 37008 Salamanca, Spain; pau-larrser@usal.es (P.A.-S.); ralonsod@usal.es (R.A.-D.); 2Institute of Biomedical Research of Salamanca (IBSAL), 37007 Salamanca, Spain; 3Laboratory of Renal Pathology and Diabetes, IIS-Fundación Jiménez Díaz, Autonoma University, 28049 Madrid, Spain; smas@quironsalud.es (S.M.-F.);; 4Diabetes and Associated Metabolic Diseases Networking Biomedical Research Centre (CIBERDEM), 28029 Madrid, Spain; 5Facultad de Ciencias Biomédicas y de la Salud, Universidad Alfonso X el Sabio (UAX), 28691 Madrid, Spain; 6Department of Nephrology and Hypertension, IIS-Fundación Jiménez Díaz, Autonoma University, 28049 Madrid, Spain; 7Fundación Renal Española, 28003 Salamanca, Spain

**Keywords:** chronic kidney disease, dialyses, protein–energy wasting, oral nutritional supplementation, intra-dialytic parenteral nutrition, nutritional assessment, quality of life, multidisciplinary intervention

## Abstract

**Background/Objectives:** Protein–energy wasting (PEW) is a common complication in patients with chronic kidney disease (CKD) receiving renal replacement therapy by dialyses. This condition is associated with higher morbidity, mortality, and poorer quality of life. The aim of this systematic review was to evaluate the effectiveness of different nutritional strategies—such as oral nutritional supplements and intra-dialytic parenteral nutrition—in improving the nutritional status of these patients. **Methods:** A systematic review was carried out in accordance with the PRISMA statement. Searches were performed in PubMed, BVS, and Scopus between January and March 2025. Randomised or controlled clinical trials published in English or Spanish, available in full text, involving adults on haemodialysis (HD) or peritoneal dialyses (PD) were included. Fourteen studies met the inclusion criteria. **Results:** The nutritional interventions assessed produced consistent benefits in biochemical markers (e.g., serum albumin), muscle mass, inflammatory indices, and perceived quality of life. Intra-dialytic supplementation and multidisciplinary management were particularly effective in patients with moderate-to-severe malnutrition. **Conclusions:** Malnutrition is frequent and clinically significant in dialyses patients. Nutritional strategies—including oral supplementation, IDPN, and personalised counselling—effectively prevent and treat PEW. Early, tailored, evidence-based, and multidisciplinary implementation could decisively improve clinical prognosis and quality of life in this population.

## 1. Introduction

Chronic kidney disease (CKD) constitutes a major global public health problem, with an estimated prevalence of 10% among the adult population [1]—particularly in older adults and in those with type 2 diabetes mellitus or arterial hypertension [2,3]. In addition to older age, type 2 diabetes, and arterial hypertension, selected congenital kidney disorders—most notably autosomal dominant polycystic kidney disease—can also contribute to the progression to advanced CKD, potentially leading to the need for dialyses [4]. Progression to advanced stages leads to an irreversible decline in renal function, frequently requiring renal replacement therapy (RRT) by haemodialysis (HD) or peritoneal dialyses (PD). HD involves extracorporeal blood purification through a semipermeable membrane, whereas PD uses the peritoneal membrane as a natural filter to remove toxins and excess fluid [5].

Patients receiving RRT have a high-risk of protein–energy wasting (PEW), affecting 18–56% of this population [6,7,8]. Recent studies from diverse regions of the world confirm that malnutrition remains highly prevalent among dialyses patients, particularly in settings with limited resources, underscoring the need for nutritional interventions tailored to the local context [9,10]. PEW is associated with increased morbidity and mortality, infections, cardiovascular events, and a marked reduction in quality of life [11,12]. Its aetiology is multifactorial, comprising iatrogenic factors such as nutrient losses during dialyses, treatment-induced inflammation, metabolic acidosis, or inadequate dialyses dose, and non-iatrogenic factors such as anorexia, taste alterations, insulin resistance, hormonal disturbances (leptin, ghrelin), and disturbances in mineral and bone metabolism (e.g., calcium–phosphate imbalance and secondary hyperparathyroidism), as well as psychosocial factors including depression or low social support [13]. In PD, additional protein losses across the peritoneum may reach 20 g per day during peritonitis episodes [14].

Inflammation plays a central role in the pathophysiology of malnutrition in CKD. Elevated pro-inflammatory cytokines (TNF-α, IL-6) promote proteolysis, loss of muscle mass, and appetite suppression [15,16], while raised C-reactive protein (CRP) correlates with poorer nutritional status and greater cardiovascular risk [17].

Sarcopenia—progressive loss of skeletal muscle mass and strength—is highly prevalent in HD patients and closely linked to PEW. Its aetiology is multifactorial, involving inflammation, hormonal changes, metabolic acidosis, and nutrient deficiencies. Sarcopenia contributes to functional impairment, reduced exercise tolerance, and diminished quality of life, and has been associated with increased morbidity and mortality. Recognising and addressing sarcopenia at an early stage is essential to preserve functional capacity and improve clinical outcomes, highlighting the importance of timely and individualised nutritional interventions in this population [18].

Early assessment of nutritional status is essential for timely management of PEW. The KDOQI guidelines recommend a 3- to 7-day dietary record—including at least one HD day—as the reference method, although food frequency questionnaires or 24 h re-calls may also be employed [19]. Validated clinical tools include the Subjective Global Assessment (SGA) [20] and the Malnutrition–Inflammation Score (MIS), which incorporates biochemical parameters such as serum albumin and total iron-binding capacity [21].

Individualised dietary counselling is the first-line nutritional intervention, enhancing adherence, identifying barriers, and tailoring intake to patient needs. When counselling alone fails to reverse PEW, oral nutritional supplements (ONS) can meet protein–energy requirements without overloading residual renal function. In cases of digestive intolerance, severe malnutrition, or poor ONS adherence, intra-dialytic parenteral nutrition (IDPN) offers an effective alternative [22].

Despite numerous studies on nutritional strategies in dialyses, methodological variability and heterogeneous interventions hinder direct comparison. This review addresses a specific evidence gap: prior syntheses have typically focused on single modalities (HD or PD) or isolated interventions and have seldom integrated recent randomised trials. We therefore map and compare the effects of oral nutritional supplements (ONS), intra-dialytic parenteral nutrition (IDPN), and selected dietary adjuncts across both HD and PD, with attention to inflammatory and functional endpoints and to implementation issues under-reported in earlier reviews. This systematic review therefore aimed to analyse the evidence on the impact of nutritional strategies—particularly ONS and IDPN—on nutritional status, body composition, biochemical markers, and quality of life in adult HD or PD patients.

## 2. Materials and Methods

### 2.1. Data Sources

This systematic review was conducted following PRISMA [23], and the research question was framed using the PICO model. Comprehensive searches of PubMed, Scopus, and the Virtual Health Library (BVS) were performed. The full PRISMA checklist is provided as Appendix A.

The search strategy combined controlled (MeSH/DeCS) and free terms in English and Spanish with the Boolean operator “AND”. Descriptors included are as follows: “Chronic kidney disease” OR “Enfermedad renal crónica”, “Malnutrition” OR “Desnutrición”, “Hemodialyses” OR “Hemodiálisis”, “Nutritional therapy” OR “Terapia nutricional”, “Dietary supplements” OR “Suplementos nutricionales”.

Searches were limited to January 2015–March 2025.

### 2.2. Eligibility Criteria

Studies were included if they met the following inclusion criteria:-Controlled or randomised clinical trials.-Adult participants (≥18 years) on HD or PD.-Articles in English or Spanish.-Free full-text availability.-Publication between 2015 and 2025.-Interventions evaluating nutritional strategies (ONS, IDPN, or combined approaches).

Exclusion criteria are as follows:-Systematic reviews, meta-analyses, or non-experimental designs.-Animal or in vitro studies.-Articles without full-text access.-Duplicates or publications in languages other than English/Spanish.-Studies unrelated to nutrition in dialyses patients.

### 2.3. Outcomes of Interest

Primary outcomes are as follows:-Changes in nutritional biomarkers (albumin, pre-albumin, transferrin).-Variations in muscle mass or body composition.-Improvement in Malnutrition–Inflammation Score (MIS).-Changes in health-related quality of life using validated questionnaires.-Secondary outcomes are as follows: Variations in inflammatory markers (CRP, IL-6) and immunological parameters.

#### Null Hypothesis

The null hypothesis of this systematic review states that there are no significant differences in nutritional, biochemical, inflammatory, or quality-of-life outcomes between dialyses patients receiving specific nutritional strategies (oral supplementation or intra-dialytic parenteral nutrition) and those receiving only conventional dietary counselling.

The review was designed to test this hypothesis and to determine whether nutritional interventions provide measurable clinical benefits compared with standard care.

### 2.4. Data Synthesis

Given the heterogeneity in study design, intervention type, duration, and outcome measures, a quantitative meta-analysis was not feasible. Results were synthesised qualitatively, comparing intervention effects by dialyses modality, intervention type, and evaluated parameters. Data were summarised in tables organised by intervention, study design, and outcomes.

## 3. Results

### 3.1. Study Selection

The systematic search yielded 1389 records: PubMed = 1153, BVS = 110, and Scopus = 126. After removing duplicates and screening titles/abstracts, 1320 records were excluded. Sixty-nine full-texts were assessed; forty-seven were excluded due to methodological issues or lack of access, leaving fourteen studies for qualitative synthesis (Figure 1).

### 3.2. Characteristics of Included Studies

The 14 selected trials (2015–2025) evaluated nutritional interventions in adult HD or PD patients. Most investigated ONS or IDPN with or without dietary counselling. Outcomes included serum albumin, muscle mass, MIS, quality of life, and inflammatory markers. Key characteristics and findings are summarised in Table 1 and Table 2.

### 3.3. Oral Nutritional Supplementation in Haemodialysis Patients

Figure 2 provides a detailed comparative overview of the nutritional interventions analysed, grouped into three categories, as follows: classic nutritional supplements, dietary adjuncts/bioactives, and probiotics. For each intervention, the figure summarises the main mechanism of action, route of administration, clinical utility, efficacy, improved outcomes, and bibliographic reference.

Seven studies evaluated ONS in HD, demonstrating significant improvements in nutritional, biochemical, body composition, and quality-of-life parameters, while effects on inflammation were less consistent.

Marini et al. [24] showed that long-term oral creatine improved fat-free mass and skeletal muscle index without altering MIS, whereas short-term supplementation reduced MIS and prevented lean mass loss [30], suggesting benefits for high-risk patients.

ONS combined with dry weight adjustment via bioelectrical impedance vector analysis (BIVA) improved serum albumin, hand-grip strength, and quality of life in the pilot study by Nieves-Anaya et al. [25]. Satirapoj et al. [26] reported that intra-dialytic ONS reduced MIS more effectively than inter-dialytic administration.

Sustained ONS for six months [6] increased serum albumin, BMI, and triceps skinfold thickness, whereas Gharib et al. [27] demonstrated improved albumin, pre-albumin, and PEW scores with reduced hs-CRP.

Amino acid replacement [28] enhanced protein intake, immunological parameters, and haematological status, while combined antioxidant supplementation with curcumin and resveratrol [29] improved muscle and bone mass and decreased ferritin.

Quality-of-life gains were documented by Nieves-Anaya et al. [25] and Gharib et al. [27], particularly in physical domains, although mental health components were unchanged.

### 3.4. Intra-Dialytic Parenteral Nutrition

Three trials investigated IDPN. Marsen et al. [31] found thrice-weekly IDPN for 16 weeks significantly increased pre-albumin, especially in moderately malnourished patients (SGA B). Kabasawa et al. [32] observed increased spontaneous protein intake and metabolic safety (fewer hypoglycaemic events) with ENEFLUID^®^ over 12 months, despite unchanged transthyretin. Kittiskulnam et al. [33] reported improved albumin, weight, MIS, and intake in patients intolerant to ONS.

IDPN therefore appears effective for moderate–severe malnutrition or when oral intake is limited, supporting recovery of nutritional status and appetite when combined with ongoing counselling.

### 3.5. Nutritional Strategies in Peritoneal Dialyses

Three PD studies assessed multidisciplinary MNT, probiotics, or whey protein supplementation. All reported beneficial effects on nutritional, inflammatory, or quality-of-life outcomes. Liang et al. [34] showed that MNT improved albumin, anaemia markers, and reduced inflammation. Pan et al. [35] conducted a randomised controlled trial in which patients received probiotic capsules containing *Bifidobacterium longum*, *Lactobacillus bulgaricus*, and *Streptococcus thermophilus* for several months. This intervention significantly reduced hs-CRP and IL-6, while improving serum albumin, mid-arm circumference, and triceps skinfold thickness. Moreover, participants reported improvements in selected domains of the SF-36 quality-of-life questionnaire, particularly those related to physical function. Sahathevan et al. [36] found that whey protein supplementation achieved recommended protein intake and improved nutritional markers in responders.

## 4. Discussion

A close examination of the studies included in this systematic review shows that several nutritional strategies—oral supplementation, intra-dialytic parenteral nutrition (IDPN), and multidisciplinary approaches—exert a beneficial effect on nutritional status, body composition, biochemical markers, and quality of life in patients undergoing dialyses. Malnutrition is highly prevalent in this population, with reported rates of 18–56% [6,7,8], arising both from the pathophysiology of chronic kidney disease itself and from dialyses-related factors such as protein losses, dietary restrictions, and persistent chronic inflammation.

Our synthesis complements current clinical guidance by integrating evidence across HD and PD and by comparing oral nutritional supplements, intra-dialytic parenteral nutrition, and selected dietary adjuncts within a single framework. This approach mirrors recommendations to individualise assessment and therapy, monitor inflammatory–nutritional phenotypes, and prioritise patient-centred outcomes [22,37]. It also addresses a persistent gap in prior reviews that typically examined single modalities or isolated interventions, thereby limiting applicability to real-world multidisciplinary care [38].

While not developed in detail here, dietary counselling is essential in the first steps of nutritional management. Oral supplementation is reserved for cases in which dietary measures prove inadequate, and IDPN should be considered only when both preceding strategies fail to improve nutritional status. This therapeutic escalation is taken for granted in routine clinical practice and constitutes the basis on which the evaluated interventions are built [19].

Within dietary counselling, one of the most widely used tools is the food record, which entails weighing and accurately recording everything consumed over three to fourteen days, including at least one weekend day. While this method is regarded as one of the most precise, it can modify patients’ eating habits during the recording period, leading to possible under-reporting of intake [39]. For a comprehensive appraisal, dietary records should be interpreted alongside paraclinical indices of systemic inflammation—such as C-reactive protein (CRP), interleukin-6 (IL-6), neutrophil-to-lymphocyte ratio (NLR), platelet-to-lymphocyte ratio (PLR), systemic immune-inflammation index (SII), serum albumin, pre-albumin, and ferritin—and composite tools like the Malnutrition–Inflammation Score (MIS) or the Patient-Generated Subjective Global Assessment (PG-SGA) [14,17,21,22]. In addition to the parameters already discussed, it is worth mentioning the Prognostic Inflammatory and Nutritional Index (PINI), which integrates markers of inflammation and nutritional status to provide a more comprehensive assessment of their balance. This index has shown prognostic value in dialyses patients and could serve as a complementary tool to the MIS or PG-SGA for improving the detection of malnutrition–inflammation and guiding more individualised interventions [40].

With regard to oral nutritional supplementation, the included studies show that this strategy improves key parameters such as serum albumin, muscle mass, and body composition. Marini et al. [30] observed an increase in muscle mass after creatine administration without significant changes in inflammatory markers—highly relevant to the management of sarcopenia, a common complication in haemodialysis (HD) characterised by loss of muscle mass and strength [41]. This phenomenon, linked to protein–energy wasting, is defined as the progressive loss of muscle and energy stores caused by an imbalance between intake and nutritional requirements [42], which justifies the recommendation to exceed 1.4 g/kg/day of protein in patients with this clinical profile [22].

Clinically, these findings support a stepwise strategy: first-line dietary counselling, followed by ONS when intake remains insufficient, and IDPN for moderate–severe PEW, and intolerance or poor adherence to oral measures [20,38]. Protein targets should align with guideline ranges (≥1.0–1.2 g/kg/day in stable HD/PD; higher intakes in PEW or catabolic states), with systematic monitoring using MIS/PG-SGA and paraclinical indices (e.g., CRP, IL-6) [22]. Given the high prevalence of sarcopenia in HD, strength-oriented interventions (e.g., creatine or amino acid supplementation) may be considered adjuncts to exercise-based programmes, using EWGSOP2 criteria to track strength and muscle mass trajectories [41].

Intra-dialytic supplementation, by contrast, has proved more effective than inter-dialytic administration [26] because it not only improves nutritional status but also facilitates therapeutic adherence, probably owing to closer medical supervision during dialyses sessions—an element that has been shown to be crucial in other chronic diseases to enhance adherence [43,44]. Nevertheless, implementing this strategy is not free of operational obstacles, such as economic cost, the need for specific infrastructure in dialyses centres [45], and sociocultural factors that may influence patient acceptance [46]. Its indication should therefore be confined to specific clinical profiles, such as patients without family support or with difficulties in maintaining autonomous feeding [11].

Several factors help explain the heterogeneity that precluded a formal meta-analysis in our review: variability in baseline nutritional phenotype (e.g., SGA A/B/C or MIS thresholds), dialyses modality (HD vs. PD) and adequacy, intervention composition and dose (calorie–protein ONS, amino acid mixtures, creatine, bioactives, probiotics), timing (intra- vs. inter-dialytic), duration (4 weeks to 12 months), adherence/supervision, and outcome selection (albumin vs. pre-albumin, MIS vs. SGA, functional indices, inflammatory markers). Divergent results in trials combining intra-dialytic nutrition with exercise or counselling further illustrate this variability, with some studies showing improvement in MIS while others report neutral effects on functional performance [26,35,47].

Implementation should consider centre-level resources (e.g., capacity for supervised intra-dialytic feeding), reimbursement, product palatability/tolerance, and patient preference, all of which influence adherence and scalability. Guidance documents emphasise that parenteral routes are reserved for patients in whom oral strategies fail or are not feasible, and that ongoing dietetic follow-up is essential to sustain benefits and minimise costs [37,45,46].

When both dietary counselling and oral supplementation are insufficient—owing to intolerance, poor adherence, or the severity of malnutrition—IDPN emerges as an effective alternative. Marsen et al. [31] and Kittiskulnam et al. [33] recorded significant improvements in pre-albumin and serum albumin, particularly when IDPN was accompanied by ongoing nutritional follow-up. This mode of support has also shown benefits in other high-risk settings, such as oncology [48] and intensive care [49], where the oral route is inadequate to meet requirements.

Among patients on peritoneal dialyses (PD), nutritional interventions likewise produced favourable effects on nutritional status and inflammatory markers. A randomised controlled trial from Asia led by Liang et al. [34] demonstrated that multidisciplinary medical nutrition therapy (MNT) reduced C-reactive protein (CRP) and improved protein and energy intake. Pan et al. [35] reported that probiotic supplementation markedly reduced CRP and IL-6, mirroring findings in inflammatory bowel diseases such as ulcerative colitis, where probiotics modulate the intestinal microbiota and decrease epithelial permeability, thereby reducing systemic inflammation [50].

Quality-of-life improvements mainly affected physical domains, underscoring the need for integrated psychological support within multidisciplinary teams [21].

Limitations include potential publication bias due to restricted database coverage, exclusion of observational studies, and small sample sizes with attrition. Most trials enrolled small cohorts, had short follow-up (<6 months), and rarely reported hard clinical endpoints such as hospitalisations, cardiovascular events, or mortality, precluding meta-analysis and limiting inferences on hospitalisations, cardiovascular events, or mortality. Additional limitations include the exclusion of older (>10 years) studies, variability in intervention protocols, and heterogeneity of outcome measures, which further complicate comparisons across studies. Taken together, these constraints substantially weaken the overall strength of the available evidence and warrant caution in interpreting the observed benefits. Emphasising these limitations highlights the gaps in current evidence and the need for adequately powered, long-term RCTs that assess clinically meaningful outcomes.

Future research should focus on standardising the design of nutritional intervention trials in dialyses patients, defining homogeneous inclusion criteria, adequate follow-up duration, and comparable outcome measures to facilitate quantitative synthesis and meta-analysis. It is particularly relevant to include inflammatory and functional biomarkers (such as CRP, IL-6, MIS, or PINI) as secondary endpoints to better characterise the malnutrition–inflammation complex and its response to therapy.

In addition, studies addressing cost-effectiveness, patient preferences, and adherence are needed to strengthen the development of patient-centred nutritional protocols. Integrating these findings into clinical guidelines (e.g., KDOQI, ESPEN) could help promote a more individualised, evidence-based practice. Finally, long-term multicentre clinical trials are required to determine whether nutritional interventions translate into improved survival, reduced hospitalisation, and enhanced overall quality of life in dialyses patients.

## 5. Conclusions

Malnutrition is frequent and clinically significant in dialyses patients. Nutritional strategies—including oral supplementation, IDPN, and personalised counselling—effectively prevent and treat PEW. Early, tailored, evidence-based, and multidisciplinary implementation could decisively improve clinical prognosis and quality of life in this population.

## Figures and Tables

**Figure 1 nutrients-17-03478-f001:**
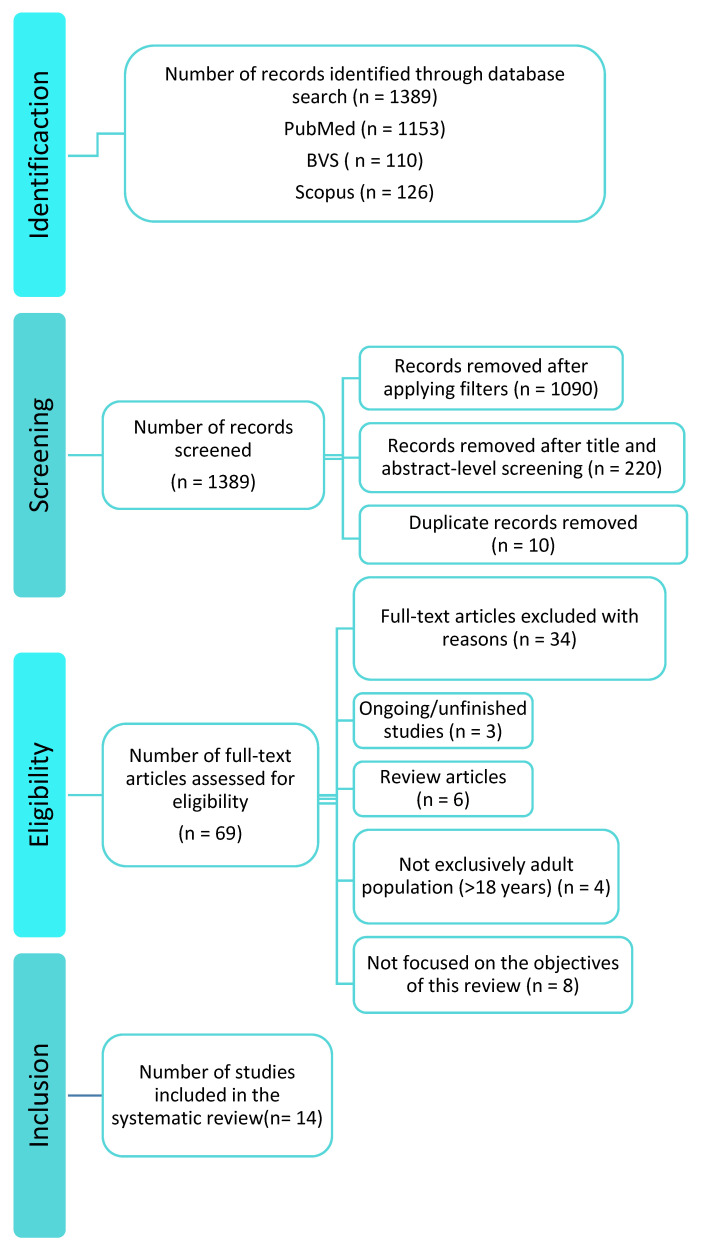
PRISMA flow diagram of the literature search.

**Figure 2 nutrients-17-03478-f002:**
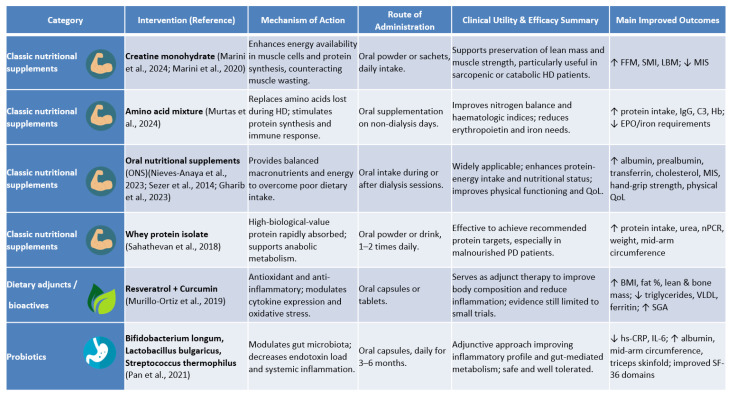
Updated comparative overview of nutritional interventions assessed in this review, grouped by type (classic nutritional supplements, dietary adjuncts/bioactives, and probiotics). Each intervention is described according to mechanism of action, route of administration, clinical utility, and evidence of efficacy [6,24,25,27,28,29,30,35,36].

**Table 1 nutrients-17-03478-t001:** Summary of the results and characteristics of the selected articles on haemodialysis.

Authors and Year	Study Design	Sample and Intervention	Methodology	Main Findings
Marini et al., 2024 [24]	Balanced randomised double-blind controlled trial	*n* = 40 HD; intervention (*n* = 21): oral sachets containing 5 g creatine monohydrate + 5 g maltodextrin daily; control (*n* = 19): 10 g maltodextrin daily	Body composition and MIS assessed at baseline, 6 months, and 12 months	↑ in fat-free mass, skeletal muscle mass index, intracellular water, total body water, and body weight in intervention group (all *p* < 0.05). MIS unchanged in both groups.
Nieves-Anaya et al., 2023 [25]	Pilot randomised study	*n* = 32 chronic HD; intervention: personalised diet + ONS (Enterex DBT) + 5.6 g protein powder; control: personalised diet only	Baseline and post-intervention assessments	Intervention reduced moderate malnutrition by 48%, whereas severe malnutrition rose by 13% in controls (*p* < 0.04). ↑ in hand-grip strength, serum albumin, transferrin, and quality of life in intervention group.
Satirapoj et al., 2024 [26]	Randomised controlled trial	*n* = 32 ESRD on HD; intra-dialytic ONS (*n* = 16; 370 kcal Once Dialyze) vs. inter-dialytic ONS (*n* = 16)	12-month follow-up	Both groups ↓ in MIS, greater reduction with intra-dialytic ONS. No significant differences in albumin, dietary intake, or anthropometry.
Sezer et al., 2014 [6]	Prospective trial	*n* = 58 HD; intervention: ONS (Nutrena) + dietary counselling (*n* = 29); control: increased dietary intake + counselling (*n* = 29)	6-month assessment	Intervention ↑ in serum albumin, LDL-cholesterol, triceps skinfold thickness, and dry weight; ↓ in EPO requirement. Controls showed ↓ in BMI and FFM and ↑ in MIS.
Gharib et al., 2023 [27]	Prospective randomised trial	*n* = 60 malnourished HD; intervention: powder ONS (NEO-MUNE) + counselling; control: counselling only	Pre-/post-comparison	Intervention ↑ in albumin, pre-albumin, cholesterol, albumin-to-surface area ratio, and PEW score; ↓ in hs-CRP (*p* ≤ 0.016).
Murtas et al., 2024 [28]	Double-blind randomised trial	*n* = 30 HD; intervention: 5.4 g amino acid mixture on inter-dialytic days; control: no supplement	3-month follow-up	Intervention ↑ in protein intake, serum IgG, complement C3, and haemoglobin; ↓ in EPO and IV iron needs (*p* < 0.05). Controls ↓ in body fat.
Murillo Ortiz et al., 2019 [29]	Randomised clinical trial	*n* = 40 CKD; intervention: 500 mg resveratrol + 500 mg curcumin daily; control: placebo	12-week assessment	Intervention ↑ in BMI, fat %, and lean and bone mass; ↓in triglycerides, VLDL, and ferritin; ↑ in SGA score. No change in oxidative stress markers.
Marini et al., 2020 [30]	Placebo-controlled randomised trial	*n* = 30 HD; intervention: creatine + maltodextrin; control: maltodextrin	4-week study	Intervention ↓ in MIS, ↑ in lean body mass (*p* < 0.05). 28.6% of controls lost lean mass vs. 14.4% in intervention.
Marsen et al., 2017 [31]	Multicentre open randomised trial	*n* = 107 HD with PEW; intervention: IDPN (aa/glucose/lipids/vitamins) 3×/week + standard counselling; control: counselling only	16 weeks	Intervention ↑ in serum pre-albumin (41% achieved ≥15% rise by week 4 vs. 20.5% controls). Greater response in moderate malnutrition (SGA B).
Kabasawa et al., 2024 [32]	Multicentre open randomised trial	*n* = 34 maintenance HD with mild–moderate malnutrition; intervention: IDPN with ENEFLUID^®^ 550 mL 3×/week; control: no IDPN	12-month follow-up	No difference in serum transthyretin; intervention ↑ in protein intake, blood urea nitrogen, and nPCR; controls ↓ in protein intake.
Kittiskulnam et al., 2022 [33]	Prospective randomised controlled trial	*n* = 38 HD with PEW intolerant to ONS; intervention: 3-in-1 fish-oil IDPN for 3 months; control: intensive dietary counselling	Baseline, 3, and 6 months	Intervention ↑ in serum albumin, body weight, MIS, and spontaneous intake; ↑ in leptin in controls.

HD = haemodialysis; PEW = protein–energy wasting; ONS = oral nutritional supplement; IDPN = intra-dialytic parenteral nutrition; MIS = malnutrition–inflammation score; EPO = erythropoietin; and nPCR = normalised protein catabolic rate.

**Table 2 nutrients-17-03478-t002:** Summary of the results and characteristics of the selected articles on peritoneal dialyses.

Authors and Year	Study Design	Sample and Intervention	Methodology	Main Findings
Liang et al., 2025 [34]	Randomised controlled trial	*n* = 81 PD; intervention: multidisciplinary medical nutrition therapy (MNT) (*n* = 41) vs. standard care (*n* = 40)	6-month follow-up	Intervention ↑ in serum albumin, calcium, iron, Hb, mid-arm circumference, triceps skinfold, hand-grip strength, and protein and energy intake; ↓ in CRP, NLR, and PLR.
Pan et al., 2021 [35]	Randomised controlled trial	*n* = 116 PD; intervention: probiotic capsules (B. *longum*, *L. bulgaricus*, *S. thermophilus*) (*n* = 58) vs. no probiotics (*n* = 58)	Pre-/post-assessment	Intervention ↓ in hs-CRP and IL-6; ↑ in serum albumin, arm circumference, and triceps skinfold; improved selected SF-36 domains.
Sahathevan et al., 2018 [36]	Multicentre open randomised trial	*n* = 126 malnourished PD; intervention: whey protein isolate sachets + dietary counselling (*n* = 65); control: counselling only (*n* = 61)	6-month follow-up	59.5% of intervention achieved adequate protein intake vs. 16.2% controls (*p* < 0.001); ↑ in serum urea, nPCR, weight, and arm circumference in responders; controls ↓ in quality of life.

PD = peritoneal dialyses; MNT = medical nutrition therapy; and nPCR = normalised protein catabolic rate.

## Data Availability

All data generated or analysed during this systematic review are included in this published article or referenced in publicly accessible databases.

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
