# Peer review of "Nutritional Strategies to Address Malnutrition in Dialyses Patients: A Systematic Review"

_nutrients, 2025, doi:10.3390/nu17213478_

Round 1

Reviewer 1 Report (Previous Reviewer 1)

Comments and Suggestions for Authors

Authors made literature search and performed a systematic review on an important issue -- nutrition in dialysis patients. Comments are as the followings.

  1. Most of the readers might not consider it appropriate to include several additives as "nutrients", such as probiotics、resveratrol、curcumin、creatine、 and fish oil.
  2. There is no hard outcome in all these included studies. 

Author Response

Reviewer #1:

Review Report #1

Comment 1: Most of the readers might not consider it appropriate to include several additives as "nutrients", such as probiotics, resveratrolcurcumincreatine and fish oil.

Response 1: We sincerely appreciate your comment. In the previous round of reviews, another reviewer specifically recommended that we distinguish between classic nutritional supplements and other products considered dietary adjuncts or bioactive additives (such as probiotics, resveratrol, curcumin, creatine, and fish oil). To address that suggestion, we prepared Table 2, which provides a visual and comparative overview of the different types of interventions, grouping them into three categories: classic nutritional supplements, dietary adjuncts/bioactives, and probiotics.

In this revised version, we have expanded the information in the table, detailing for each intervention its mechanism of action, route of administration, clinical utility, and reported outcomes, in order to further clarify the conceptual and functional differences between these categories. We believe this updated presentation helps readers clearly identify which interventions represent true nutritional supplementation and which should be interpreted as complementary or metabolic modulators.

Comment 2: There is no hard outcome in all these included studies. 

Response 2: We sincerely appreciate your observation. We fully agree that most of the included studies did not assess hard clinical outcomes (such as hospitalization, cardiovascular events, or mortality), which indeed represents an important limitation in the current literature on nutritional interventions in dialysis patients.

However, this limitation was explicitly acknowledged in the Discussion section, where we noted that the small sample sizes and short follow-up periods of the available trials precluded a meta-analysis and warrant caution when interpreting the observed benefits. In this regard, we share the reviewer’s concern and emphasize that one of the key conclusions of our work is precisely the need for future, adequately powered clinical trials that include these clinically meaningful endpoints.

Reviewer 2 Report (Previous Reviewer 2)

Comments and Suggestions for Authors

I think that the authors apported the right corrections.

Author Response

We sincerely thank the reviewer for their positive feedback and for acknowledging that the corrections made have adequately addressed the previous concerns.

Reviewer 3 Report (Previous Reviewer 3)

Comments and Suggestions for Authors

See enclosed pdf

Author Response

Reviewer #3:

Review Report #3

Comments 1: The authors should state the null hypotgesus

Response 2: We appreciate the reviewer’s comment. A new subsection (2.3.1 Null hypothesis) has been added in the Materials and Methods section, explicitly stating the null hypothesis of the review. It clarifies that no significant differences are expected between patients receiving nutritional strategies (oral supplementation or intradialytic parenteral nutrition) and those following standard dietary counselling alone, thereby defining the comparative framework of the study.

Comments 2: The authors could also insert an image with the differences in action, means of aplication, utility and efficacy of proposed suplementation

Response 2: We appreciate the reviewer’s suggestion. Figure 2 has been updated to expand the comparative information on the nutritional interventions analysed. The revised version now includes, in addition to the mechanism of action and main outcomes, the routes of administration and a summary of the clinical utility and evidence of efficacy for each type of intervention (classic nutritional supplements, dietary adjuncts/bioactives, and probiotics). This update provides a more comprehensive visual overview of the differences in action, mode of application, and effectiveness of the reviewed strategies, thus addressing the reviewer’s recommendation.

Comments 3: The authors should also discuss the utility of certain indexes to assess inflammation and malnutrition status in dialisis patients, such as PINI to better evaluate these oatients. I suggest: Cordos, M.; Martu, M.-A.; Vlad, C.-E.; Toma, V.; Ciubotaru, A.D.; Badescu, M.C.; Goriuc, A.; Foia, L. Early Detection of Inflammation and Malnutrition and Prediction of Acute Events in Hemodialysis Patients through PINI (Prognostic Inflammatory and Nutritional Index). Diagnostics 2024, 14, 1273. https://doi.org/10.3390/diagnostics14121273

Response 3: We thank the reviewer for this valuable suggestion. The Discussion section has been expanded to include a specific reference to the Prognostic Inflammatory and Nutritional Index (PINI), as proposed by the reviewer. Its prognostic usefulness in dialysis patients is highlighted, as well as its potential role as a complementary tool to established instruments such as the MIS or PG-SGA to enhance the assessment of malnutrition-inflammation and guide more individualised nutritional management.

Comment 4: The authors could elaborate on future research, it is an interesting section that is insuficently covered.

Comments 5: The authors should state the clinical importance and future usage in the elaboration of patient centered protocols and clinical guidelines

Responses 4 and 5: We thank the reviewer for the valuable comments. The final part of the Discussion has been expanded to address both remarks jointly, elaborating on future research priorities and the clinical implications of the findings. The revised text highlights the need to standardise study designs in nutrition and dialysis, incorporate inflammatory and functional biomarkers (CRP, IL-6, MIS, PINI) into outcome assessments, and evaluate cost-effectiveness, adherence, and patient preferences. It also emphasises the relevance of these aspects for the development of patient-centred protocols and clinical guidelines, aiming to translate evidence into more individualised and evidence-based practice. This addition reinforces both the scientific and clinical significance of the review.

Reviewer 4 Report (New Reviewer)

Comments and Suggestions for Authors

Review Report

Title: Nutritional Strategies to Address Malnutrition in Dialysis Patients: A Systematic Review

General Assessment

This manuscript addresses an important and clinically relevant topic. Malnutrition and protein–energy wasting in dialysis patients remain major challenges in nephrology, and synthesizing the evidence on nutritional interventions is timely. The review is well-structured, generally clear, and follows PRISMA guidelines. The inclusion of both haemodialysis and peritoneal dialysis patients, as well as the comparison of oral nutritional supplements and intradialytic parenteral nutrition, strengthens the work. The conclusions are clinically meaningful and consistent with current guidelines.

However, some methodological clarifications and editorial improvements are needed before the paper is suitable for publication.

Major Weaknesses

  1. Search transparency: The methods section lacks a detailed PubMed search string and does not indicate PROSPERO registration or provision of a PRISMA checklist. Including these would improve reproducibility and transparency.
  2. Results presentation: Table 1 is dense and difficult to read. Splitting it by dialysis modality (HD vs PD) or simplifying presentation would improve clarity.
  3. Discussion redundancy: Dietary counselling as the first-line intervention is repeated several times almost verbatim, which could be streamlined.
  4. Limitations of evidence: The limitations of included RCTs (small sample sizes, short follow-up, lack of hard clinical outcomes such as mortality or hospitalization) are noted but not emphasized sufficiently.

Minor Weaknesses

  1. Typographical errors (e.g., “moder-ate” on line 31) should be corrected.
  2. Some sentences are overly long and complex; the manuscript would benefit from stylistic polishing and simplification.
  3. Figure 2 could be better linked to the corresponding outcomes for improved interpretability.
  4. References are up-to-date but somewhat Eurocentric; more global context could be added.

Author Response

Review Report #4

General Assessment

This manuscript addresses an important and clinically relevant topic. Malnutrition and protein–energy wasting in dialysis patients remain major challenges in nephrology, and synthesizing the evidence on nutritional interventions is timely. The review is well-structured, generally clear, and follows PRISMA guidelines. The inclusion of both haemodialysis and peritoneal dialysis patients, as well as the comparison of oral nutritional supplements and intradialytic parenteral nutrition, strengthens the work. The conclusions are clinically meaningful and consistent with current guidelines.

However, some methodological clarifications and editorial improvements are needed before the paper is suitable for publication.

Response: We thank the reviewer for their positive and constructive comments. We are pleased that they find our manuscript relevant and well-structured. We have incorporated their suggestions to improve the clarity ad rigor of the work.

Major Weaknesses

Comment 1: Search transparency: The methods section lacks a detailed PubMed search string and does not indicate PROSPERO registration or provision of a PRISMA checklist. Including these would improve reproducibility and transparency.

Response 1: We thank the reviewer for this suggestion. We have included the PRISMA checklist as supplementary material to improve transparency and reproducibility.

Comment 2: Results presentation: Table 1 is dense and difficult to read. Splitting it by dialysis modality (HD vs PD) or simplifying presentation would improve clarity.

Response 2: We have revised Table 1 and split it into two separate tables according to dialysis modality (HD vs. PD) to improve clarity a readability.

Comment 3: Discussion redundancy: Dietary counselling as the first-line intervention is repeated several times almost verbatim, which could be streamlined.

Response 3: We thank the reviewer for pointing out the redundancy in the discussion. We have revised the text and simplified the repeated mentions of dietary counselling as a first-line intervention and oral supplementation, thus improving the clarity and conciseness of the manuscript.

Comment 4: Limitations of evidence: The limitations of included RCTs (small sample sizes, short follow-up, lack of hard clinical outcomes such as mortality or hospitalization) are noted but not emphasized sufficiently.

Response 4: We have revised the Limitations section to emphasize small sample size, short follow-up, and the lack of hard clinical outcomes such as mortality and hospitalizations, highlighting how these factors limit the strength of the available evidence.

Minor Weaknesses

Comment 5: Typographical errors (e.g., “moder-ate” on line 31) should be corrected.

Response 5: We thank the reviewer for pointing out the typographical errors. These have been corrected throughout the manuscript.

Comment 6: Some sentences are overly long and complex; the manuscript would benefit from stylistic polishing and simplification.

Response 6: We have revised the manuscript to simplify overly long and complex sentences and improve overall readability.

Comment 7: Figure 2 could be better linked to the corresponding outcomes for improved interpretability.

Response 7: We thank the reviewer for this helpful comment. To improve the interpretability of Figure 2 and its linkage with the corresponding outcomes, we have expanded and revised the figure. The updated version now details, for each intervention, the mechanism of action, route of administration, clinical utility, efficacy, improved outcomes, and bibliographic reference, allowing a clearer visual connection between the interventions and the results presented in sections 3.3–3.5. In addition, the accompanying explanatory text has been rewritten to reflect these enhancements and to ensure consistency in terminology and order with the Results section.

Comment 8: References are up-to-date but somewhat Eurocentric; more global context could be added.

Response 8: Thank you for your valuable suggestion. We have addressed your comment by incorporating additional references that offer a more global perspective beyond the European context in the introduction, and we have highlighted the Asian origin of a randomized controlled trial included in the review.

Round 2

Reviewer 1 Report (Previous Reviewer 1)

Comments and Suggestions for Authors

My previous statement still holds.

1. Authors' description puts emphasis on PEW very much, and readers would not understand why resveratrol、probiotics、curcumin ... might have anything to do with PEW. Is there any evidence to prove that those additives could improve PEW?

2. The "Conclusion" in the Abstract -- "Their individualised implementation could substantially improve clinical outcomes and quality of life in this vulnerable population." is very confirmative. The present article did not provide evidence on this.

Reviewer 3 Report (Previous Reviewer 3)

Comments and Suggestions for Authors

the manuscript has been improved

Reviewer 4 Report (New Reviewer)

Comments and Suggestions for Authors

None

This manuscript is a resubmission of an earlier submission. The following is a list of the peer review reports and author responses from that submission.

Round 1

Reviewer 1 Report

Comments and Suggestions for Authors

Authors made a systemic review on nutritional strategies for dialysis patients. Their conclusion approved the effect of either oral or parenteral nutritional supplement. Comments are as the followings.

  1. The major drawback of the present analysis is the great difference in the content of supplement in the included studies. Some of them are not actually "nutritional" supplement, especially resveratrol、curcumin、and probiotics.
  2.  The number of patients is small. Outcome is not assessed (certainly limited by the included studies). 

Author Response

Comments 1: The major drawback of the present analysis is the great difference in the content of supplement in the included studies. Some of them are not actually "nutritional" supplement, especially resveratrol、curcumin、and probiotics.

Response 1: In response to the reviewer’s comment, we have added in section 3.3. Oral nutritional supplementation in haemodialysis patients a new figure (Figure 2) that classifies the interventions analysed into three categories: classic nutritional supplements, dietary adjuncts/bioactives, and probiotics. This makes it explicit that some of the products evaluated (e.g., resveratrol, curcumin, or probiotics) are not conventional nutritional supplements, but rather interventions with complementary mechanisms of action and objectives. This graphical representation, which also addresses a suggestion from another reviewer, includes for each intervention its main mechanism, improved parameters, and corresponding reference, thereby facilitating comparison between strategy types and interpretation of the findings. The number of patients is small. Outcome is not assessed (certainly limited by the included studies). 

Comments 2: The number of patients is small. Outcome is not assessed (certainly limited by the included studies). 

Response 2: We agree with the reviewer’s observation. To address this, we have reinforced the Limitations and discussion sections by explicitly noting the small sample size of most included trials, the rare reporting of hard clinical endpoints, and the resulting impossibility of conducting a meta-analysis. We have also specified that these limitations restrict inferences regarding hospitalisations, cardiovascular events, and mortality. In the Future research section, we now highlight the need for adequately powered, multicentre RCTs with ≥12-month follow-up to evaluate patient-centred outcomes, clinical events, and survival.

Revised text – Discussion: Several factors help explain the heterogeneity that precluded a formal meta-analysis in our review: variability in baseline nutritional phenotype (e.g., SGA A/B/C or MIS thresholds), dialysis modality (HD vs PD) and adequacy, intervention composition and dose (calorie–protein ONS, amino-acid mixtures, creatine, bioactives, probiotics), timing (intra- vs interdialytic), duration (4 weeks to 12 months), adherence/supervision, and outcome selection (albumin vs prealbumin, MIS vs SGA, functional indices, inflammatory markers). Divergent results in trials combining intradialytic nutrition with exercise or counselling further illustrate this variability, with some studies showing improvement in MIS while others report neutral effects on functional performance

Revised text – Limitations:
Limitations include potential publication bias due to restricted database coverage, exclusion of observational studies, small sample sizes with attrition. Most trials enrolled small cohorts and rarely reported hard clinical endpoints, precluding meta-analysis and limiting inferences on hospitalisations, cardiovascular events, or mortality. Additional limitations include the exclusion of older (>10 years) studies and limited follow-up (<6 months), which hampers assessment of long-term effects and hard clinical outcomes. Taken together, these constraints substantially weaken the overall strength of the available evidence and warrant caution in interpreting the observed benefits.

Revised text – Future research:
Priorities include personalised education programmes to enhance ONS adherence, early malnutrition detection using tools such as MIS or PG-SGA, and effective multidisciplinary coordination. Adequately powered, multicentre RCTs with ≥12-month follow-up are required to evaluate patient-centred outcomes, clinical events, and survival. Such studies are essential to confirm the benefits observed so far and provide a more robust evidence base to guide clinical practice.

Reviewer 2 Report

Comments and Suggestions for Authors

the paper is very interesting and fit for the publication in the journal but I have some observations

In the Introduction, the authors should insert also congenital diseases as Polycistic Kindney

At page 2, line 52, it is useful to insert alterations calcium metabolism

At page 8,line 196, I suggest to specify the type of Probiotics

At page 9, line 232, the authors speak about Sarcopenia but I suggest to insert this concept with relative explanations in the introduction

Author Response

Comments 1: In the Introduction, the authors should insert also congenital diseases as Polycistic Kindney

Response 1: We have revised the Introduction to explicitly mention congenital kidney diseases (e.g., autosomal dominant polycystic kidney disease) as contributors to progression to advanced CKD requiring dialysis.

Revised text – Introduction: “In addition to older age, type 2 diabetes and arterial hypertension, selected congenital kidney disorders—most notably autosomal dominant polycystic kidney disease—also contribute to progression to advanced CKD requiring dialysis.”

Comments 2: At page 2, line 52, it is useful to insert alterations calcium metabolism

Response 2: We integrate in the text the calcium-phosphorus metabolism disturbances“Disturbances in mineral and bone metabolism (e.g., calcium–phosphate imbalance and secondary hyperparathyroidism) further interact with inflammation and muscle catabolism, potentially worsening nutritional status.”

Comments 3: At page 8,line 196, I suggest to specify the type of Probiotics

Response 3: We specify in the text the strains for Pam et al.

Revised text - “Pan et al. [31] conducted a randomised controlled trial in which patients received probiotic capsules containing Bifidobacterium longum, Lactobacillus bulgaricus and Streptococcus thermophilus for several months. This intervention significantly reduced hs-CRP and IL-6, while improving serum albumin, mid-arm circumference, and triceps skinfold thickness. Moreover, participants reported improvements in selected domains of the SF-36 quality-of-life questionnaire, particularly those related to physical functioning”

Comments 4: At page 9, line 232, the authors speak about Sarcopenia but I suggest to insert this concept with relative explanations in the introduction

Response 4: We have relocated and expanded the definition and clinical relevance of sarcopenia to the Introduction, explicitly linking it to protein–energy wasting (PEW) and to functional outcomes.

Revised text -“Sarcopenia—defined as a progressive and generalised loss of skeletal muscle mass and strength—is highly prevalent in patients undergoing haemodialysis and is closely linked to protein–energy wasting. Its aetiology is multifactorial, involving inflammation, hormonal changes, metabolic acidosis, and nutrient deficiencies. Sarcopenia contributes to functional impairment, reduced exercise tolerance, and diminished quality of life, and has been associated with increased morbidity and mortality. Recognising and addressing sarcopenia at an early stage is essential to preserve functional capacity and improve clinical outcomes, highlighting the importance of timely and individualised nutritional interventions in this population”

Reviewer 3 Report

Comments and Suggestions for Authors

please see enclosed pdf

Author Response

Response 1: We have emphasized the specific gap addressed by this review, integrating evidence from both hemodialysis and peritoneal dialysis, oral nutritional supplements (ONS) and intradialytic parenteral nutrition (IDPN), as well as dietary adjuvants, with particular attention to inflammatory and functional markers, along with implementation issues.

Revised text . “This review addresses a specific evidence gap: prior syntheses have typically focused on single modalities (HD or PD) or isolated interventions and have seldom integrated recent randomised trials. We therefore map and compare the effects of oral nutritional supplements (ONS), intradialytic parenteral nutrition (IDPN) and selected dietary adjuncts across both HD and PD, with attention to inflammatory and functional endpoints and to implementation issues under-reported in earlier reviews.”

Response 2: Following the reviewer’s suggestion, we have included in section 3.3. Oral nutritional supplementation in haemodialysis patients a figure (Figure 2) detailing the specific characteristics of the different nutritional interventions evaluated, organised into three categories: classic nutritional supplements, dietary adjuncts/bioactives, and probiotics. The figure includes, for each intervention, its main mechanism, improved parameters, and bibliographic reference, allowing for a clearer and more comparative visualisation of the findings. This figure also addresses another reviewer’s observation regarding the heterogeneity of the supplements included, highlighting those that are not conventional nutritional supplements.

Response 3: In line with the reviewer’s suggestion, we have substantially expanded the Discussion section to integrate not only the literature already cited in the manuscript but also relevant clinical guidelines (KDOQI, ESPEN) and additional recent references. Specific paragraphs have been added to:

  1. Place the findings in the context of current recommendations and highlight the evidence gap addressed by this review.
  2. Deepen the clinical implications, outlining a stepwise approach (dietary counselling- oral nutritional supplementation- intradialytic parenteral nutrition) and nutritional targets according to guideline ranges.
  3. Explain the sources of heterogeneity that precluded meta-analysis (differences in patient populations, dialysis modalities, type and duration of intervention, assessed parameters, etc.).
  4. Address practical implementation considerations (resources, adherence, tolerance, cost).

These additions broaden the discussion, improve its coherence with the available evidence, and provide a more comprehensive interpretation of the results and their implications for clinical practice.

Response 4: In the Discussion section, within the paragraph addressing the use of dietary records, we have incorporated a sentence underscoring the importance of complementing these data with paraclinical inflammation indices and composite tools to achieve a more comprehensive assessment.

Revised text - “For a comprehensive appraisal, dietary records should be interpreted alongside paraclinical indices of systemic inflammation—such as C-reactive protein (CRP), interleukin-6 (IL-6), neutrophil-to-lymphocyte ratio (NLR), platelet-to-lymphocyte ratio (PLR), serum albumin, prealbumin and ferritin—and composite tools like the Malnutrition–Inflammation Score (MIS) or the Patient-Generated Subjective Global Assessment (PG-SGA)“

Round 2

Reviewer 1 Report

Comments and Suggestions for Authors

Authors response and explanation have not changed the drawback.

Reviewer 3 Report

Comments and Suggestions for Authors

The manuscript is good